# Learning with Alignments: Tackling the Inter- and Intra-domain Shifts for Cross-multidomain Facial Expression Recognition

Author(s) Names(s): Anonymous ACM MM Submission

## ABSTRACT

Facial Expression Recognition (FER) holds significant importance in human-computer interactions. Existing cross-domain FER methods often transfer knowledge solely from a single labeled source domain to an unlabeled target domain, neglecting the comprehensive information across multiple sources. Nevertheless, cross-multidomain FER (CMFER) is very challenging for (i) the inherent inter-domain shifts across multiple domains and (ii) the intra-domain shifts stemming from the ambiguous expressions and low inter-class distinctions. In this paper, we propose a novel Learning with Alignments CMFER framework, named LA-CMFER, to handle both inter- and intra-domain shifts. Specifically, LA-CMFER is constructed with a global branch and a local branch to extract features from the full images and local subtle expressions, respectively. Based on this, LA-CMFER presents a dual-level inter-domain alignment method to force the model to prioritize hard-to-align samples in knowledge transfer at a sample level while gradually generating a well-clustered feature space with the guidance of class attributes at a cluster level, thus narrowing the inter-domain shifts. To address the intra-domain shifts, LA-CMFER introduces a multi-view intra-domain alignment method with a multi-view clustering consistency constraint where a prediction similarity matrix is built to pursue consistency between the global and local views, thus refining pseudo labels and eliminating latent noise. Extensive experiments on six benchmark datasets have validated the superiority of our LA-CMFER.

## CCS CONCEPTS

• Computing methodologies→Artificial intelligence; • Human-centered computing→Human computer interaction (HCI)

## KEYWORDS

Cross-multidomain adaptation, Inter- and intra-domain shifts, Facial expression recognition, Domain alignments

## 1 Introduction

Facial Expression Recognition (FER) endeavors to discern human expressions and emotional states, playing a pivotal role in human-computer interactions [1]. Nowadays, FER has gained notable promotion thanks to diverse deep learning (DL) algorithms [2, 3, 4, 5, 46, 47, 48] and well-annotated FER datasets [6, 7, 8]. However, these FER works [9, 10] typically operate under the assumption that both training and testing samples come from the same dataset (domain) and inherently share an identical data distribution. In practice, their classification accuracy often drops sharply due to the great discrepancy in data distribution (i.e., inter-domain shifts) [11, 12] when applied in different scenes, making them incapable of tackling the cross-domain problem settings.

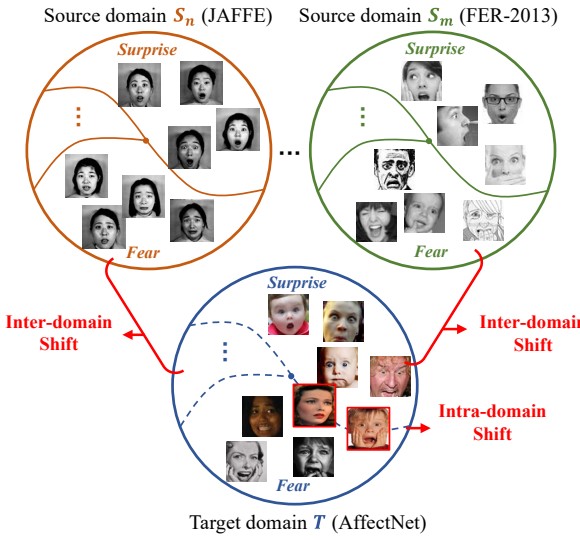

**Fig. 1. Illustration to the problem setting of CMFER task where both inter-domain shifts (different data distributions) and intra-domain shifts (ambiguous expressions and low inter-class distinctions near the target decision boundaries) exist.**

To alleviate the inter-domain shifts, unsupervised Cross-Domain FER (CDFER) [13, 14, 49] has been introduced, aiming to extract domain-invariant features from a single labeled source domain to accurately classify the samples in an unlabeled target domain. Various techniques, such as adversarial learning [11, 15, 16] and metric learning [12, 17, 18], have been explored to reduce the distribution disparity between the two domains. However, these methods primarily focus on leveraging data information from a single source, while overlooking the valuable facial knowledge buried in multiple source domains. In reality, there are multiple labeled source datasets collected with diverse conditions, such as different acquisition environments, ethnic characteristics, etc. Leveraging these sources can significantly increase the number of training samples and further provide more comprehensive and discriminative feature representations of facial expressions, thus contributing to better generalization ability. Subsequently, unsupervised Cross-Multidomain FER (CMFER) [19] is motivated to effectively transfer richer facial knowledge from multiple labeled source domains to the unlabeled target domain.

A simple solution to CMFER tasks is to regard different sources as a single source and directly apply CDFER methods. Regretfully, as the number of sources increases, these approaches often exhibit sub-optimal performance or even degradation owing to the growing

complexity of data distributions [19, 20]. Therefore, developing a well-designed CMFER method is quite imperative. Delving deeply into CMFER, there are two main challenges. The first one is the above-mentioned *inter-domain shifts across different domains* due to their inherent data distribution difference. Specifically, as shown in Fig.1, with different shooting conditions, AffectNet [6] (i.e., $T$) contains both gray and color images, while FER-2013 [21] (i.e., $S_m$) only includes gray images. Besides, JAFFE [22] (i.e., $S_n$) has notable racial characteristics where all the facial images come from Japanese women. These severe inter-domain shifts disrupt the effective extraction of domain-invariant facial expression representations and hinder smooth knowledge transfer from source domains to the target domain. The second challenge for CMFER is the *intra-domain shifts that occur within the target domain*. As shown in Fig.1, the subtle differences and intrinsic ambiguities in human expressions (marked by red borders) may lead to lower inter-class distinctions. Therefore, the intra-domain shifts are likely to cause the model to produce uncertain category predictions and acquire inaccurate semantic information, severely damaging the prediction robustness.

In this paper, we propose a novel **L**earning with **A**lignments for **Cross-M**ultidomain **F**acial **E**xpression **R**ecognition (LA-CMFER) framework to simultaneously tackle the inter- and intra-domain shifts. In LA-CMFER, we consider both essential global and local features for facial expressions and employ a shared dual-branch structure across multiple source domains and the target domain. Then, to address the inter-domain shifts, a dual-level (i.e., sample-level and cluster-level) inter-domain alignment is developed to encourage the model to pay more attention to the hard-to-align samples with higher uncertainty and generate a more well-clustered feature space based on the predicted labels. As for the intra-domain shifts, a multi-view intra-domain alignment is devised with a muti-view clustering consistency constraint to form a discriminative target feature space, and a multi-view voting scheme to gain the high-quality pseudo labels for an effective information interaction between the local and global branches, respectively. Both inter- and intra-domain alignments act with united strength to comprehensively extract discriminative facial representations from multiple source domains and promote the classification in the target domain. The main contributions of this paper are three-fold:

- We present a Learning with Alignments for Cross-Multidomain Facial Expression Recognition (LA-CMFER) framework to tackle both inter- and intra-domain shifts and utilize beneficial knowledge from multiple source domains to address the challenging CMFER task.
- To mitigate the inter-domain shift, we introduce a dual-level inter-domain alignment method that prioritizes hard-to-align samples and fosters a well-clustered feature space. As for the intra-domain shift, we design a muti-view clustering consistency constraint between the global and local branches to gain a concentrated target feature space for alleviating the interference of noisy samples.
- Extensive experiments on six commonly used FER benchmark datasets verify the superior performance of our method compared to other state-of-the-art approaches.

## 2 Related Works

### 2.1 Cross-domain FER

CDFER follows the fundamental principles of Unsupervised Domain Adaptation (UDA) [23], aiming to apply the classifier trained with a single labeled source domain to classify samples in an unlabeled target domain. Numerous studies have focused on CDFER tasks, which can be categorized into two groups: metric-based approaches vs. adversarial-based approaches. Concretely, metric-based approaches [12, 13 17, 18, 24] construct proper distance functions to extract more discriminative facial features for reducing cross-domain distribution variation. For instance, Ni et al. [24] combined metric learning with dictionary learning to alleviate the transfer facial expression recognition issue. Zhang et al. [13] presented a local-global discriminative subspace transfer learning (LGDSTL) method where a local-global graph is used as distance metric. Besides, adversarial-based approaches [11, 15, 16, 25] mine the domain-invariant facial features with adversarial training. To illustrate, Xie et al. [15] embedded a graph representation propagation with adversarial learning and presented an adversarial graph representation adaptation framework. Considering the important local facial features, Ji et al. [11] proposed a region attention-enhanced CDFER approach to emphasize the local features for minimizing domain discrepancies. Despite their promising results, these methods deal with only one single source and are not competitive in addressing the complex data distributions encountered in cross-multidomain scenarios.

### 2.2 Multi-source Domain Adaptation

Multi-source Domain Adaptation (MDA) [26] holds the assumption that data can be gathered from diverse source domains with different distributions, which is a more practical but challenging task compared to single-source domain adaptation. Pioneer studies [27, 28, 50] have theoretically confirmed that the target distribution can be represented as a weighted combination of source distributions. To further ensure effective adaptation, the key lies in overcoming the inter-domain shifts caused by diverse data distributions across domains. Along with this philosophy, Zhao et al. [29] used adversarial training to align the target and source distributions. Zhu et al. [30] developed an MDA framework with multiple domain-specific classifiers to mine the domain-invariant features and use a maximum mean discrepancy (MMD) loss [31] to ease the inter-domain shift. Li et al. [32] devised a feature filtration network to selectively align features across domains.

Unlike existing MDA works in natural image classification, the CMFER tasks are more difficult. Besides the inter-domain shifts, the subtle differences between facial expressions also bring about intra-domain shifts, e.g., low inter-class distinctions in the decision boundaries. In the emerging CMFER, only Liu et al. [19] proposed a domain-uncertain mutual learning (DUML) network based on adversarial learning to consider both inter-domain and intra-domain uncertainty. In this paper, we present the Learning with Alignments framework with dual-level inter-domain alignment and multi-view intra-domain alignment to adeptly address two kinds of domain shifts with a simpler and lighter architecture.

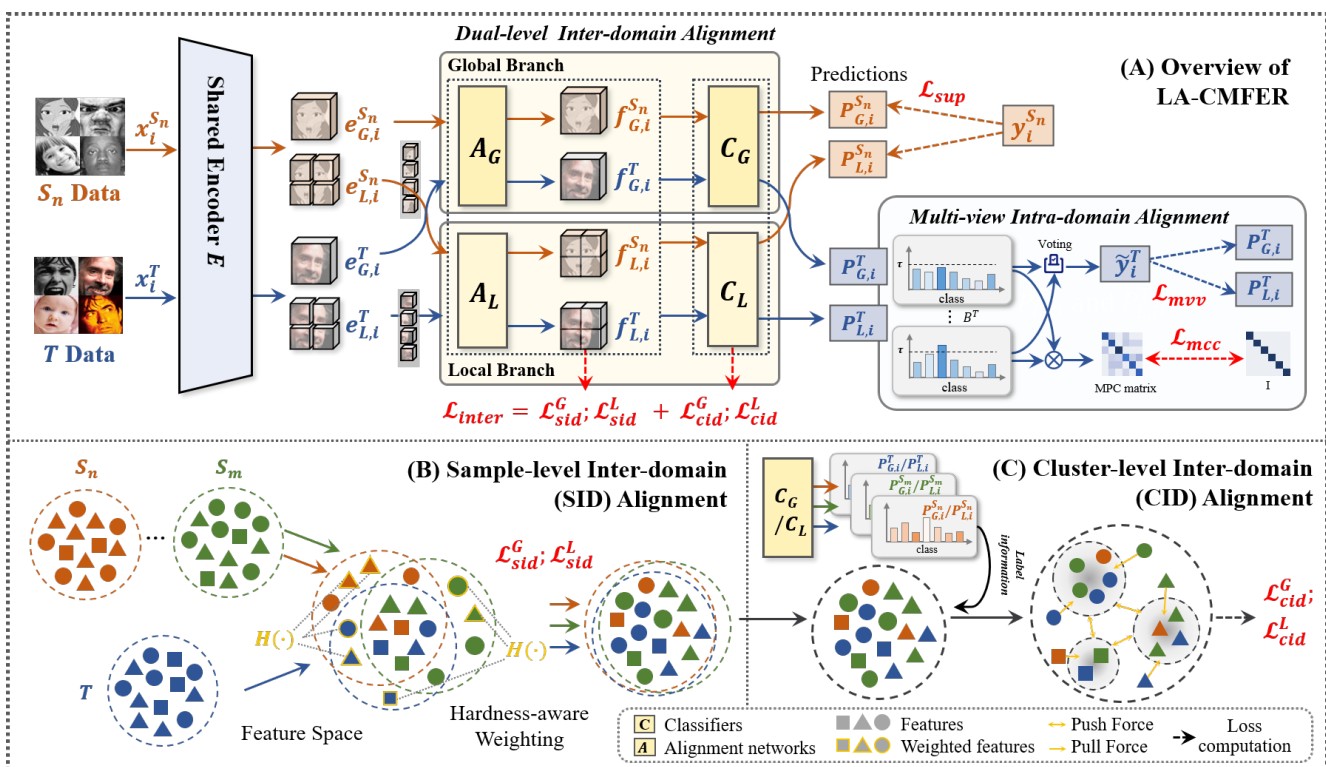

**Fig. 2. Overview of our proposed LA-CDFER framework with a dual-level inter-domain alignment and a multi-view intra-domain alignment. Our framework receives samples with labels from $|S_n|$ sources and tries to classify the target samples in $\mathcal{D}^T$.**

## 3 Methodology

### 3.1 Preliminaries and Framework Overview

In the problem definition of CMFER, there are $N$ labeled source domains $\{S_n\}_{n=1}^N$ and an unlabeled target domain $T$, with a total of $K$ shared facial expression classes. Concretely, $S_n$ denotes the $n$-th FER dataset $\mathcal{D}^{S_n} = \{(x_i^{s_n}, y_i^{s_n})\}_{i=1}^{|S_n|}$ where $x_i^{s_n}$ is the $i$-th labeled source image and $y_i^{s_n} \in \{0,1\}^K$ represents its one-hot real label. Correspondingly, the target domain $T$ contains a single unlabeled dataset $\mathcal{D}^T = \{x_i^T\}_{i=1}^{|T|}$, where $x_i^T$ denotes the $i$-th unlabeled target image. CMFER tries to train a deep model with $\mathcal{D}^T$ and $\{\mathcal{D}^{S_n}\}_{n=1}^N$ to accurately predict the expression labels of $x^T$ in $\mathcal{D}^T$.

The overview of our LA-CDFER framework is illustrated in Fig. 2, which includes a shared encoder $E$ with both global and local branches. Specifically, the global/local branch is equipped with respective global/local feature alignment network $A_G / A_L$ and feature classifier $C_G/C_L$. Notably, $A_G$ and $A_L$ hold different model architectures to respectively mine the expression knowledge from global and local views. In the training stage, the shared encoder $E$ first encodes the source image $x_i^{s_n}$ from the $n$-th source batch ($B^{s_n}$) into compact embedding $e_{G,i}^{s_n}$. For the global branch, $e_{G,i}^{s_n}$ is fed into $A_G$ to produce the global feature $f_{G,i}^{s_n}$ and $C_G$ is then utilized to form the final prediction $P_{G,i}^{s_n}$. Meanwhile, we produce a local embedding

$e_{L,i}^{s_n}$ by cropping and concatenating $e_{G,i}^{s_n}$ in a grid manner. After the processing of $A_L$ and $C_L$, the local feature $f_{L,i}^{s_n}$ and prediction $P_{L,i}^{s_n}$ are obtained. The same process is applied to the target image $x_i^T$, yielding its global/local feature $f_{G,i}^T / f_{L,i}^T$ as well as prediction $P_{G,i}^T/P_{L,i}^T$. To deal with the inter-domain shifts, LA-CMFER uses a dual-level inter-domain alignment with loss $\mathcal{L}_{inter}$ to separately align global and local source domain features with the target ones at both sample- and cluster-levels. For the intra-domain shifts, given $P_{G,i}^T$ and $P_{L,i}^T$, a multi-view consistency constraint loss $\mathcal{L}_{mcc}$ is utilized to ensure consistency between the global and local views while eliminating latent noise. Besides, a multi-view voting scheme is devised after the $\mathcal{L}_{mcc}$ to generate high-quality pseudo labels $\tilde{y}_i^T$ and supervise the predictions of the two branches in turn, facilitating the knowledge interaction of the two branches.

### 3.2 Global and Local Branches

Given the notable importance of both global and local features, LA-CMFER adopts a dual-branch architecture. As shown in Fig. 2, fed with the source image $x_i^{s_n}$, encoder $E$ first obtains its compact embedding $e_{G,i}^{s_n}$, which is then processed into a global feature $f_{G,i}^{s_n}$ using the alignment network $A_G$. To capture finer details essential for recognizing subtle differences in full-face images, $e_{G,i}^{s_n}$ is further subdivided into four attentional regions: top-left ($e_{tl,i}^{s_n}$), top-right ($e_{tr,i}^{s_n}$), bottom-left ($e_{bl,i}^{s_n}$), and bottom-right ($e_{br,i}^{s_n}$). These embeddings $e_{L,i}^{s_n} = \{e_{tl,i}^{s_n}, e_{tr,i}^{s_n}, e_{bl,i}^{s_n}, e_{br,i}^{s_n}\}$ separately undergo

processing by $A_L$ and are then concatenated to form a local feature $f_{L,i}^{S_n}$. Finally, the classifiers $C_G$ and $C_L$ generate predictions $P_{G,i}^{S_n} = C_G(f_{G,i}^{S_n})$ and $P_{L,i}^{S_n} = C_L(f_{L,i}^{S_n})$. Similarly, when processing the target image $x_i^T$, the same procedure is followed, resulting in two predictions from the two branches, i.e., $P_{G,i}^T = C_G(A_G(E(x_i^T)))$ and $P_{L,i}^T = C_L(A_L(E(x_i^T)))$. These two branches make predictions from the complementary global-local views for both source and target samples, thus better detecting the beneficial expression knowledge.

## 3.3 Dual-level Inter-domain Alignment

To tackle the inter-domain shifts between source and target distributions, LA-CMFER presents the dual-level (sample- and cluster-level) inter-domain alignment to comprehensively extract the domain-invariant facial features. This is gained by prioritizing hard-to-align samples with higher uncertainty and grouping samples based on their category information.

**Sample-level Inter-Domain Alignment**. Existing CDFER works usually use the discrepancy-based constraints (e.g., MMD [30, 31, 52], triplet loss [4], and adversarial loss [11, 25]) to minimize the inter-domain shift. Concretely, given the source distribution $\mathcal{P}^{S_n}(x, y)$ and target distribution $\mathcal{P}^T(x, y)$, the traditional MMD loss [31] can be formulated as follows:

$$\mathcal{L}_{MMD}(\mathcal{P}^{S_n}, \mathcal{P}^T) = \widehat{D}_{\varkappa}(\mathcal{P}^{S_n}, \mathcal{P}^T) = \left\| \bar{\phi}^{S_n} - \bar{\phi}^T \right\|_{\varkappa}^2, \quad (1)$$

$$\bar{\phi}^a = \frac{1}{|B^a|} \sum_{i=1}^{|B^a|} \phi_i^a, \quad (2)$$

where $\phi_i^a$ denotes the average feature mappings of input samples within the batch $B^a$ to the reproducing Kernel Hilbert Space $\varkappa$ and $a$ represents the target domain ($T$) or the $n$-th source ($S_n$).

Despite its satisfactory performance, traditional MMD usually averages the distances among samples with uniform weights, thus neglecting the varying importance of different samples in the alignment. In the CMFER task, some hard-to-align samples, exhibiting higher uncertainty and located near decision boundaries, may offer greater insights into the domain alignment and knowledge transfer, thus warranting increased attention. Thus, we propose the sample-level inter-domain (SID) alignment with a hardness-aware weighting function. Specifically, given an image $x_i^a$ and its prediction $P_{M,i}^a$ with $J$ probability values $\{P_{M,i}^{a,j}\}_{j=1}^J$, where $M = G \ or \ L$ denotes different branches and $J$ denotes the number of classes, we use a measurement term $\Omega_M(\cdot)$ to adaptively evaluate the instantaneous hardness of $x_i^a$ in the current iteration:

$$\Omega_M(x_i^a) = \sqrt{\sum_{j=1}^J \mathbb{1}[P_{M,i}^{a,j} \neq Max(P_{M,i}^a) \ or \ j \neq rce] (P_{M,i}^{a,j})^2}, \quad (3)$$

where $\mathbb{1}[\cdot]$ is a binary indicator and $Max(\cdot)$ represents the maximum function. From both local and global views, $\Omega_M(\cdot)$ aims to exclude the maximum probability in $x_i^T$ or set the real class element (rce) to 0 in $x_i^{S_n}$ and then evaluate the $L2$-norm of the remaining elements. This prioritizes uncertain target sample $x_i^T$ or source sample $x_i^{S_n}$ by assigning them with higher weights. Subsequently, $\Omega_M(x_i^a)$ is normalized in $B^a$ to determine the relative weights $H_M(x_i^a)$ to facilitate batch-wise optimization:

$$H_M(x_i^a) = \frac{\Omega_M(x_i^a)}{\sum_{x_k^a \in B^a} \Omega_M(x_k^a)}. \quad (4)$$

Finally, $H_M(\cdot)$ is utilized to modify the traditional MMD loss as our sample-level inter-domain alignment loss $\mathcal{L}_{sid}^M$:

$$\mathcal{L}_{sid}^M(\mathcal{P}^{S_n}, \mathcal{P}^T) = \left\| \bar{\phi}_H^{S_n} - \bar{\phi}_H^T \right\|_{\varkappa}^2, \quad (5)$$

$$\bar{\phi}_H^a = \frac{1}{|B^a|} \sum_{i=1}^{|B^a|} H_M(x_i^a) \cdot \phi_i^a. \quad (6)$$

Notably, $H(\cdot)$ can be seamlessly integrated into other discrepancy-based constraints in a plug-and-play way for providing useful perceptions about sample importance.

**Cluster-level Inter-domain Alignment**. Existing discrepancy-based inter-domain alignments [30, 31] may blindly align samples with different classes closer, potentially impeding the model from learning discriminative features. Thus, we introduce a cluster-level inter-domain alignment to better group samples based on their categories. We use the real label $y_i^{S_n}=k$ and pseudo label $\hat{y}_i^T=\hat{d}$ to denote the category attributes of the source sample $x_i^{S_n}$ and target sample $x_i^T$, respectively. Then, we minimize the distances between source-target samples of the same category ($\hat{d} = k$) and maximize them for samples of different categories ($\hat{d} \neq k$). This process is summarized as a cluster-level inter-domain alignment loss $\mathcal{L}_{cid}$:

$$\mathcal{L}_{cid}^M(\mathcal{P}^{S_n}, \mathcal{P}^T) = \left\| \bar{\phi}_{H,k}^{S_n} - \bar{\phi}_{H,\hat{d}=k}^T \right\|_{\varkappa}^2 - \left\| \bar{\phi}_{H,k}^{S_n} - \bar{\phi}_{H,\hat{d}\neq k}^T \right\|_{\varkappa}^2, \quad (7)$$

$$\bar{\phi}_{H,k}^{S_n} = \frac{1}{|B_k^{S_n}|} \sum_{i=1}^{|B_k^{S_n}|} \bar{\phi}_{H,i}^{S_n}, \ \bar{\phi}_{H,\hat{d}}^T = \frac{1}{|B_{\hat{d}}^T|} \sum_{i=1}^{|B_{\hat{d}}^T|} \bar{\phi}_{H,i}^T, \quad (8)$$

where the hardness-aware function $H(\cdot)$ is retained to dynamically balance the importance of each sample. As $\mathcal{L}_{cid}$ optimizes, we can achieve a more fine-grained alignment to foster an intra-class convergence while promoting inter-class divergence.

Finally, the dual-level inter-domain alignment is performed in both global and local branches at the same time, thus deriving the following inter-domain loss $\mathcal{L}_{inter}$:

$$\mathcal{L}_{inter}(\mathcal{P}^{S_n}, \mathcal{P}^T) = \sum_{n=1}^N \left[ (\mathcal{L}_{sid}^G + \mathcal{L}_{sid}^L) + \lambda(\mathcal{L}_{cid}^G + \mathcal{L}_{cid}^L) \right], \quad (9)$$

where $\lambda$ is a hyper-parameter for balancing these two terms. Notably, for all domains, we include only those samples whose maximum prediction probability exceeds a predefined reliability threshold $\epsilon$, thus preventing unreliable knowledge transfer from excessively uncertain samples.

## 3.4 Multi-view Intra-domain Alignment

**Multi-view Clustering Consistency Constraint:** The intra-domain shifts within the FER target domain are typically characterized by noisy samples near the decision boundary, which may lead to a dispersed target feature space, thereby introducing prediction bias. To tackle this, we propose a novel multi-view clustering consistency constraint, aiming to shape a discriminative and concentrated target feature space. In this space, clusters with the same class predicted by classifiers of different branches should remain similar, while those of different classes should be distinct. Specifically, given the predictions $P_{G,B}^T$ and $P_{L,B}^T$ of all target samples in the target batch $B^T$ from the global and local branches, we build a Multi-view Prediction Consistency (MPC) matrix to evaluate the consistency between these predictions with the following formulation:

$$MPC = (P_{G,B}^T)' \otimes P_{L,B}^T. \quad (10)$$

**Algorithm 1:** Training procedure of our LA-CMFER.

1: **Input:** $|S_n|$ source datasets $\mathcal{D}^{S_n} = \{(x_i^{S_n}, y_i^{S_n})\}_{i=1}^{|S_n|}$ with their corresponding expression labels and the single target dataset $\mathcal{D}^T = \{x_i^T\}_{i=1}^{|T|}$ without labels.

2: **Initialize:** Initialize the network parameters: $\theta_E$ for $E$, $\theta_{AG}$ for $A_G$, $\theta_{AL}$ for $A_L$, $\theta_{CG}$ for $C_G$, $\theta_{CL}$ for $C_L$.

3: **for** $j = 0$ to $iter\_total$ **do**

5:    Take out a batch of samples $B^{S_n}$ and $B^T$ from $(x_i^{S_n}, y_i^{S_n})$ and $\mathcal{D}^T$

6:    Compute $P_{G,i}^{S_n} = C_G(A_G(E(x_i^{S_n})))$, $P_{L,i}^{S_n} = C_L(A_L(E(x_i^{S_n})))$ $P_{G,i}^T = C_G(A_G(E(x_i^T)))$, and $P_{L,i}^T = C_L(A_L(E(x_i^T)))$

7:    Compute the relative hardness weights $H_M(x_i^a)$ where $M = G$ or $L$, $a = S_n$ or $T$ with Eq. (4)

8:    Compute the sample-level inter-domain alignment loss $\mathcal{L}_{sid}^M$ where $M = G$ or $L$ with Eq. (5)

9:    Compute the cluster-level inter-domain alignment loss $\mathcal{L}_{cid}^M$ where $M = G$ or $L$ with Eq. (7)

10:   Compute the total inter-domain loss $\mathcal{L}_{inter}$ with Eq. (9)

11:   Compute the multi-view clustering consistency loss $\mathcal{L}_{mcc}$ with Eq. (11)

12:   **if** $\mathbb{1}[MV(P_{G,i}^T, P_{L,i}^T)]$

13:      Compute $\tilde{y}_i^T$ and multi-view voting loss $\mathcal{L}_{mvv}$ with Eq. (12)

14:   Compute the supervised loss $\mathcal{L}_{sup}$ with Eq. (13)

15:   Update $\{\theta_E, \theta_{AG}, \theta_{AL}, \theta_{CG}, \theta_{CL}\}$ by minimizing Eq. (14)

16: **end for**

---

Drawing on the cluster assumption [33], we consider target samples in $B^T$ as a specific class (i.e., class $c$) and their corresponding prediction probabilities in $B^T$ as a batch-wise cluster representation for class $c$, which is captured by the $c$-th column in $P_{G,B}^T$ or $P_{L,B}^T$. To reach the intra-domain alignment, we enhance the consistency among cluster representations for identical class assignments (i.e., higher diagonal values in MPC) and inconsistency among them with different class assignments (i.e., lower off-diagonal values in MPC). This process can be represented as the following multi-view clustering consistency loss $\mathcal{L}_{mcc}$:

$$\mathcal{L}_{mcc} = \frac{1}{2c}\big(S(\|MPC\| - I) + S(\|MPC'\| - I)\big), \quad (11)$$

where $S(\cdot)$ calculates the sum of absolute values across all matrix elements, $I \in \mathbb{R}^{K \times K}$ represents an identity matrix, and $\|\cdot\|$ denotes a normalization function [34]. Minimizing $\mathcal{L}_{mcc}$ facilitates the well-separated clusters of different expressions in the target feature space and encourages consistent predictions from dual branches.

**Multi-view Voting Scheme:** Considering the high-quality pseudo labels can provide more accurate guidance, we propose a multi-view scheme with two voting conditions to select them: (i) global and local branch makes consistent predictions from different views, i.e., $argmax(P_{G,i}^T) = argmax(P_{L,i}^T)$; (ii) at least one branch achieves higher confidence than a threshold $\tau$, i.e., $Max(P_{G,i}^T) > \tau$ or $Max(P_{L,i}^T) > \tau$. When the two points are all satisfied, the high-quality pseudo-label $\tilde{y}_i^T$ is selected. Additional supervision is then provided to $P_{G,i}^T$ and $P_{L,i}^T$ using a multi-view voting loss $\mathcal{L}_{mvv}$:

$$\mathcal{L}_{mvv} = \frac{1}{|B^T|}\sum_{i=1}^{|B^T|}\mathbb{1}[MV(P_{G,i}^T, P_{L,i}^T)] \times [CE(P_{G,i}^T, \tilde{y}_i^T) + CE(P_{L,i}^T, \tilde{y}_i^T)], \quad (12)$$

where $MV(\cdot)$ is a function to determine whether the two voting conditions are simultaneously met. By incorporating $\mathcal{L}_{mcc}$ and $\mathcal{L}_{mvv}$, LA-CMFER is guided to achieve a meaningful knowledge

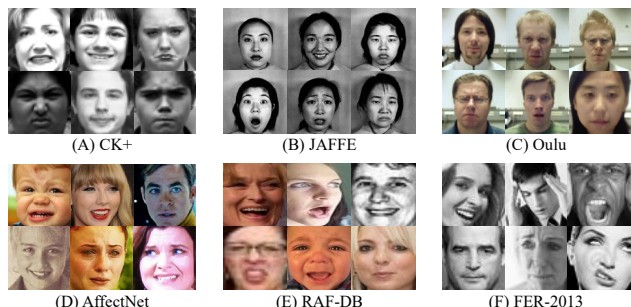

(A) CK+       (B) JAFFE       (C) Oulu

(D) AffectNet       (E) RAF-DB       (F) FER-2013

**Fig. 3. Examples from the six FER datasets.**

interaction between the two branches, promoting mutual enhancement and facilitating the development of a more robust target feature space.

## 3.5    Training Objective

The whole training objective contains five parts: (1) supervised loss $\mathcal{L}_{sup}$ for labeled source samples, (2) inter-domain loss $\mathcal{L}_{inter}$, (3) multi-view clustering consistency loss $\mathcal{L}_{mcc}$, and (4) multi-view voting loss $\mathcal{L}_{mvv}$ for unlabeled target samples.

The supervised loss $\mathcal{L}_{sup}$ can be represented as follows:

$$\mathcal{L}_{sup} = \frac{1}{N}\sum_{n=1}^{N}[CE(P_{L,i}^{S_n}, y_i^{S_n}) + CE(P_{G,i}^{S_n}, y_i^{S_n})], \quad (13)$$

where $CE(\cdot)$ denotes the cross-entropy loss.

Finally, the training objective of LA-CMFER is formulated as:

$$\mathcal{L}_{total} = \mathcal{L}_{sup} + \alpha\mathcal{L}_{inter} + \beta\mathcal{L}_{mcc} + \gamma\mathcal{L}_{mvv}, \quad (14)$$

where $\alpha$, $\beta$, and $\gamma$ are three weighted hyper-parameters.

The training stage of LA-CMFER is summarized in Algorithm 1. In the inference stage, we take the prediction with a higher probability from the two branches as the final result [51].

## 4    Experiments and Result Analysis

### 4.1    Experimental Setup

**Datasets.** To verify the performance of the proposed method, we follow the experiment settings in [19] and involve six commonly used FER datasets, including three lab-controlled datasets CK+ [7], JAFFE [22], and Oulu-CASIA [35] and three Internet-collected large-scale field datasets AffectNet [6], RAF-DB [8], and FER-2013 [21]. Specifically, **CK+** provides 593 annotated video samples from 123 subjects and uses the last three frames with six basic emotions (i.e., anger, disgust, fear, surprise, happy, and sad) and neutral emotions, finally gaining a total of 1,236 images. **JAFFE** includes 213 facial expression images from 10 Japanese women. **Oulu-CASIA (Oulu)** captures 2988 face images by choosing the last three frames with normal lighting for basic expressions and the first frame for neutral expressions. **AffectNet** stands as the largest facial expression dataset to date. Following [19], we curate up to 5,000 facial images for each emotion category, resulting in a total of 33,793 training images and 3,500 testing images. **RAF-DB** is a real-world FER dataset that has 12,271 and 3,068 images for training and testing, respectively. **FER-2013**, compiled via the Google image search engine, contains 35,887

**Table 1: Comparisons (%) with SOTAs across multidomain FER datasets.**

| Protocols | Methods | Venues | →JAFFE | →RAF-DB | →CK+ | →Oulu | →AffectNet | →FER-2013 | Avg |
|---|---|---|---|---|---|---|---|---|---|
| Single Best | Source-Only | - | 59.15 | 58.74 | 77.58 | 54.21 | 49.31 | 51.46 | 55.67 |
| | LGDSTL [13] | TAC 2022 | - | 45.13 | 66.67 | - | - | 33.88 | - |
| | RANDA [11] | TKDE 2021 | - | 62.48 | 88.71 | - | 52.34 | - | - |
| | DMSRL [43] | TMM 2022 | 68.54 | - | 88.51 | 64.38 | 52.54 | 58.63 | - |
| | AGLRLS [14] | TMM 2024 | 61.97 | - | 87.60 | - | - | **60.68** | - |
| | DAN [4] | ICML 2015 | 59.62 | 59.35 | 76.46 | 56.35 | 51.57 | 52.72 | 59.35 |
| | DANN [41] | JMLR 2016 | 61.03 | 60.53 | 77.43 | 54.23 | 50.29 | 54.67 | 59.67 |
| | ADDA [42] | CVPR 2017 | 62.91 | 61.47 | 80.02 | 57.50 | 51.89 | 53.30 | 61.18 |
| Source Combine | Source-Only | - | 53.52 | 60.10 | 77.83 | 58.44 | 41.86 | 42.24 | 55.67 |
| | DAN [4] | ICML 2015 | 54.75 | 62.32 | 80.34 | 59.48 | 47.46 | 50.52 | 59.14 |
| | DANN [41] | JMLR 2016 | 60.56 | 64.21 | 79.85 | 60.78 | 49.86 | 51.46 | 60.95 |
| | ADDA [42] | CVPR 2017 | 61.50 | 64.77 | 83.17 | 62.86 | 46.23 | 51.99 | 61.75 |
| | AGLRLS [14] | TMM 2024 | 60.56 | 65.32 | 83.80 | 61.91 | 49.54 | 53.27 | 62.40 |
| Multi-Source | MDAN [29] | NIPS 2018 | 60.09 | 63.36 | 79.29 | 62.24 | 48.23 | 51.16 | 60.73 |
| | M³SDA [44] | ICCV 2019 | 51.17 | 65.41 | 72.57 | 55.26 | 46.37 | 47.62 | 56.40 |
| | MJD [45] | PR 2024 | 61.03 | 68.29 | 80.48 | 63.45 | 49.83 | 52.43 | 62.59 |
| | DUML [19] | MM 2023 | 69.95 | 73.24 | 88.57 | 65.60 | 52.46 | 56.56 | 67.73 |
| | LA-CMFER (ours) | 2024 | **70.42** | **77.86** | **90.48** | **66.50** | **53.26** | 57.40 | **69.32** |

images of different facial expressions and uses 28,709 images for training and 3,589 images for testing. Intuitively, some samples from each dataset are given in Fig. 3.

For label settings, we select the subset with six basic and neutral emotion labels from each dataset, and the training and testing sets of the lab-controlled datasets are the same. In our experiments, each dataset is sequentially regarded as a domain and we utilize the symbol '→A' to represent that dataset A is the target domain while other datasets serve as source domains.

**Implementation Details.** We conduct all our experiments on two GeForce RTX 3090 GPUs with the PyTorch toolbox. Same as [19], face images are first detected and aligned by the RetinaFace [36] and further resized to 224 ×224. We use the conv1~conv3 layers of ResNet-18 pre-trained on ImageNet [37] as the shared encoder $E$. *The details of the global and local branches are in Supplementary Materials.* For all datasets, we train the whole framework for 20,000 iterations with a learning rate of 0.01 and a batch size of 128. The setting of the optimizer and learning schedule are kept the same with [38]. In our experiments, we maintain a fixed random seed over 3 runs and report the mean results. The hyper-parameter $\lambda$ is empirically set as 0.02. For the hyper-parameters in Eq. (14), based on our trial studies, we set $\alpha$ as 0.4 for FER-2013 and AffectNet and 0.1 for the rest. Besides, we set $\beta$ as 0.5 and $\gamma$ as 0.1. The reliability threshold $\epsilon$ is set as 0.4. To select reliable pseudo labels, $\tau$ is set as 0.9 [39, 40].

## 4.2 Comparison Experiments

To verify the effectiveness of the proposed method, we compare our method with state-of-the-art (SOTA) Single-source DA methods (SDA) and MDA methods. For SDA methods, two protocols are adopted: (1) Single Best, which reports the best result among all source domains, and (2) Source Combine, which naively combines all source domains and then uses the SDA methods. Specifically, Source-Only refers to directly transferring the model trained in source domains to the target domain. We select (i) the traditional SDA methods: DAN [4], DANN [41], ADDA [42], and

CD-FER method AGLRLS [14] with both protocols. (ii) The CD-FER methods: LGDSTL [13], DMSRL [43], and RANDA [11] are chosen as additional comparison methods for Single Best. For MDA methods, we choose (iii) the typical MDA methods MDAN [29], M³SDA [44], and MJD [45]; and (iv) the latest CMFER method DUML [19]. To ensure a fair comparison, the results of these methods are obtained either from their respective papers or by reimplemented using their released code.

The comparison results on different FER datasets are reported in Tab.1 where our LA-CMFER gains the best average accuracy of 69.32% and largely surpasses the second-best DUML by 1.59%. In particular, all Source Combine methods are relatively unpromising, as the inter-domain shifts impair effective knowledge transfer. On the '→CK+' and '→FER-2013' tasks, the SOTA CDFER method AGLRLS experiences considerable performance degradation (i.e., 62.40% average accuracy) when adapting from single-source to multi-source scenarios. This underscores the impracticality of solely relying on the Source Combine strategy to address CMFER tasks. Besides, by tackling the inter-domain shifts, MDA methods gain better performance where MJD obtains 62.59% average accuracy and reaches 68.29% and 63.45% on the '→RAF-DB' and '→Oulu' task, respectively. As a well-designed SOTA CMFER method, DUML achieves the second-best overall performance. Compared to it, our LA-CMFER still keeps its leading accuracy. Especially, on the challenging '→RAF-DB' task, LA-CMFER remarkably surpasses DUML by 4.62%. These experimental results have verified the superior performance of our LA-CMFER.

## 4.3 Analytical Experiments

**Contributions of Key Components:** To study the contributions of key components in our LA-CMFER, we progressively conduct ablation experiments on all FER datasets. For clarity, 'Baseline' means directly transferring the dual-branch model trained in the source domains to the target domain with only supervised loss $\mathcal{L}_{sup}$. 'MMD' indicates using the traditional MMD loss to assist the network to alleviate the inter-domain shift. The experimental

**Table 2: Ablation studies (%) on the contributions of our key components.**

| Variants | →J | →R | →C | →O | →A | →F | Avg |
|---|---|---|---|---|---|---|---|
| (A) Baseline | 53.52 | 60.05 | 84.76 | 58.40 | 41.81 | 42.35 | 56.82 |
| (B) w. MMD | 61.03 | 73.31 | 89.05 | 63.92 | 50.40 | 52.94 | 65.11 |
| (C) w. $H(\cdot)$ | 66.67 | 75.88 | 89.52 | 63.59 | 51.03 | 55.48 | 67.03 |
| (D) w. $\mathcal{L}_{inter}$ | 68.08 | 76.38 | 89.04 | 64.92 | 51.43 | 56.28 | 67.69 |
| (E) w. $\mathcal{L}_{mcc}$ | 69.48 | 77.37 | 89.71 | 65.93 | 52.69 | 56.84 | 68.67 |
| (F) w. $\mathcal{L}_{mvv}$ | **70.42** | **77.86** | **90.48** | **66.50** | **53.26** | **57.40** | **69.32** |

**Table 3: Comparison results (%) of different inter-domain alignment strategies.**

| Strategy | →J | →R | →C | →O | →A | →F | Avg |
|---|---|---|---|---|---|---|---|
| Adv variant | 68.54 | 77.76 | 89.81 | 61.85 | 47.99 | 52.02 | 66.33 |
| Ours | **70.42** | **77.86** | **90.48** | **66.50** | **53.26** | **57.40** | **69.32** |

**Table 4: Quantitative analysis of features distances achieved by Adv variant, DUML, and our LA-CMFER, respectively.**

| Methods | Adv variant | | DUML | | Ours | |
|---|---|---|---|---|---|---|
| Domain | Sources | Target | Sources | Target | Sources | Target |
| Accuracy (%↑) | - | 66.33 | - | 67.73 | - | **69.32** |
| (A) Intra-L2 (↓) | 0.85 | 1.40 | 0.78 | 1.35 | **0.69** | **1.08** |
| (B) Intra-var (↓) | 0.02 | 0.02 | 0.02 | 0.02 | 0.02 | 0.02 |
| (C) Inter-L2 (↑) | 1.32 | 1.02 | 1.48 | 1.06 | **1.50** | **1.32** |
| (D) $r$ (↑) | 1.55 | 0.73 | 1.90 | 0.78 | **2.17** | **1.22** |

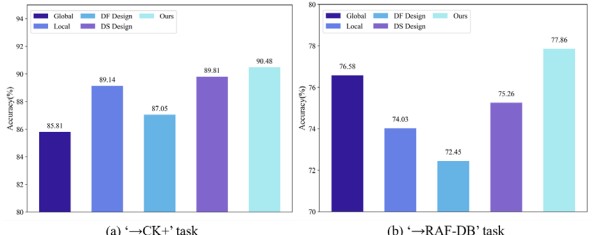

**Fig. 4. Experimental results about different global or local architectures on the (a) '→CK+' and (b) '→RAF-DB' tasks.**

alignment effectiveness. Thus, we adopt several distance metrics, involving (A) intra-class L2 distance (intra-L2), (B) intra-class variance (intra-var), (C) inter-class L2 distance (inter-L2), and (D) distance ratio $r(r=$ inter-L2/intra-L2). The comparisons among the Adv variant, DUML [19], and our LA-CMFER are shown in Tab.4. As seen, compared to both the Adv variant and DUML, our LA-CMFER achieves the least intra-class distances (0.69 for sources and 1.08 for target) and the largest inter-class distances (1.50 for sources and 1.32 for target). Such remarkable superiorities validate that our LA-CMFER can effectively promote inter-class variability and intra-class compactness, thus reaching the best accuracy (69.32%).

**Analysis of the Global-Local Branch Architecture:** To capture both the global knowledge from the full image and local expression details, LA-CMFER is built with a global-local branch architecture. To study its superiority, we compared five different architectures: (i) only global branch (denoted as 'Global'), (ii) only local branch (denoted as 'Local'), (iii) domain-fusion design where the features from the global and local branches are concatenated and fed into a single classifier (denoted as 'DF Design'), (iv) domain-specific design [19, 30] where a global branch, as well as a local branch, are adopted to each source domain and the final results come from a weighted of all classifiers (denoted as 'DS Design'), and (v) our global-local branch with only a global classifier and a local classifier. The results are summarized in Fig. 4. DF Design and DS Design sometimes perform worse than Global or Local. This could be due to the distortion of important local and global information, as well as interference from the differences in distribution among multiple sources and their insufficient information interactions. Overall, compared to DS Design, like DUML, our design global-local branch architecture presents a simpler and lighter architecture with no additional discriminators and domain-specific modules, and can better handle the CMFER task adeptly and effectively transfer knowledge across domains.

**Impact of Different Cross-view Consistency Constraints:** In the intra-domain alignment, the consistency loss enforces the identical predictions of the global and local branches. Here, we explore the performance disparities by using four multi-view consistency constraints in our LA-CMFER network: (i) Kullback-Leibler (KL) divergence, (ii) L1 distance (), (iii) Mean Squared Error (MSE), and (iv) our proposed multi-view clustering consistency loss ($\mathcal{L}_{mcc}$). Experimental results on each task are displayed in Fig. 5. As seen, KL divergence achieves the second-best accuracies on almost all tasks while our $\mathcal{L}_{mcc}$ surpasses it largely. This verifies that $\mathcal{L}_{mcc}$

results are shown in Tab.2 where we use the initial letters of the dataset as its reference for the space limitations. As seen, each constraint loss positively contributes to performance enhancements in most cases. With MMD, (B) achieves 8.29% mean promotion by narrowing the inter-domain shift within the dual-branch framework. Compared to (B), the sample-level inter-domain alignment with hardness-aware weighting function $H(\cdot)$, i.e., (C), further enhances the accuracy from 65.11% to 67.03%. Besides, with the additional cluster-level alignment, (D) gains notable promotions on '→ JAFFE' (↑1.41%) and '→Oulu' (↑1.33%) tasks, respectively. Then, with $\mathcal{L}_{mcc}$ and $\mathcal{L}_{mvv}$ in the multi-view intra-domain alignment, we obtain mean accuracies of 68.67% and 69.32%, respectively.

**Impact of Different Inter-domain Alignment Strategies:** To explore the impact of different inter-domain alignment strategies, we compare our dual-level inter-domain alignment with the widely used adversarial training strategy [19]. To accomplish adversarial training, rather than employing our dual-level inter-domain alignment, we use two discriminators following our global and local branches to distinguish whether the features come from the source domain or the target domain (denoted as 'Adv variant'). As seen in Tab.3, our dual-level inter-domain alignment achieves enhancements on all tasks, particularly for the '→O' (↑4.65%), '→ A' (↑5.27 %), and '→F' (↑5.38%) tasks. These results strongly reveal that, unlike adversarial training, which only aligns multiple domains at a feature level, our method can prioritize hard samples with abundant knowledge and further leverage label information, thus enabling a more fine-grained alignment with higher accuracy.

Further, in the domain alignment, the distances among the data distributions of multiple domains can give a direct evaluation of the

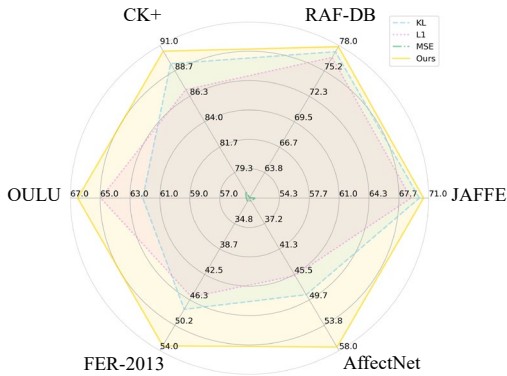

**Fig. 5. Experimental results about different cross-view consistency constraints on each task.**

**Table 6: Analytical experiment of different multi-view pseudo labeling strategies on the '→CK+' and '→RAF-DB' tasks.**

| Strategy | →CK+ | →RAF-DB | Avg |
|---|---|---|---|
| SPL | 85.43 | 74.40 | 79.92 |
| Ours | **90.48** | **77.86** | **84.17 (↑4.25)** |

better fosters consistency between two branches, thus decreasing the prediction uncertainty and promoting intra-domain alignments.
**Impact of the Different Multi-view Pseudo Labeling Strategies:** In the intra-domain alignment, LA-CMFER designs a multi-view voting scheme to provide additional supervision ($\mathcal{L}_{mvv}$) with the high-quality pseudo labels which are consistently generated by two branches and at least one branch achieves a higher confidence than the threshold $\tau$. Here, we compare our scheme with a separate pseudo labeling (SPL) strategy where each branch is independently constrained with high-confidence pseudo labels within the branch. The experimental results are given in Tab.6. On the '→ CK+' and '→RAF-DB' tasks, our multi-view voting scheme gains 5.05% and 3.46% notable accuracy enhancements, respectively, thus verifying its superiority for global-local knowledge interaction.
**Hyper-parameter Sensitivity Tests:** In Eq. (14), we utilize hyper-parameters $\alpha$, $\beta$, and $\gamma$ to balance our four losses. Here, we perform hyper-parameter sensitivity tests to determine the optimal values of the three parameters, and the results are shown in Fig. 6. For the '→CK+' task, when $\alpha$, $\beta$, and $\gamma$ are respectively set as 0.1, 0.5, and 0.1, we obtain the highest accuracies. And when $\alpha$, $\beta$, and $\gamma$ are respectively set as 0.4, 0.5, and 0.1, the best results are gained on the '→FER-2013' task. Thus, we set $\alpha$ as 0.4 for the hard dataset FER-2013 and AffectNet while 0.1 for the rest. Besides, we set $\beta$ and $\gamma$ as 0.5 and 0.1, respectively. The hyper-parameter $\lambda$ in Eq. (9) balances the sample- and cluster-level inter-domain alignment, and we try to find its optimal value. Since the cluster-level alignment only finetunes the feature space with label information, we select the value of $\lambda$ from {0.01, 0.02, 0.04, 0.06, 0.08, 0.10}. As seen in Tab. 7, our model is robust to $\lambda$ and when $\lambda$ is set as 0.02, the best accuracies are gained.
**Feature visualizations:** To assess the transferability of our model, we employ t-SNE visualizations to visualize the feature embeddings of different models on the '→CK+' task. As illustrated in Fig. 7, the feature embeddings of the Baseline present a

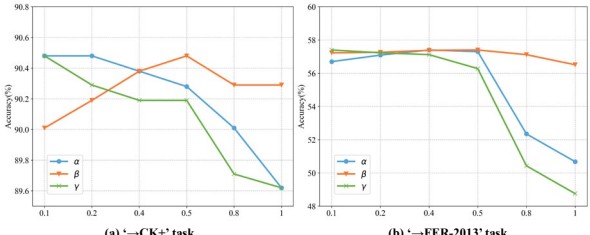

**Fig. 6. Experimental results about sensitivity test of hyper-parameters $\alpha$, $\beta$, and $\gamma$.**

**Table 7: Experimental results about different selections of hyper-parameter $\lambda$ on the '→CK+' and '→FER-2013' tasks.**

| | Hyper-parameter $\lambda$ | | | | | |
|---|---|---|---|---|---|---|
| | 0.01 | 0.02 | 0.04 | 0.06 | 0.08 | 0.10 |
| Accuracy (%) on '→CK+' | 90.19 | **90.48** | 90.38 | 90.29 | 90.19 | 89.81 |
| Accuracy (%) on '→FER-2013' | 57.12 | **57.40** | 57.26 | 57.12 | 56.78 | 56.26 |

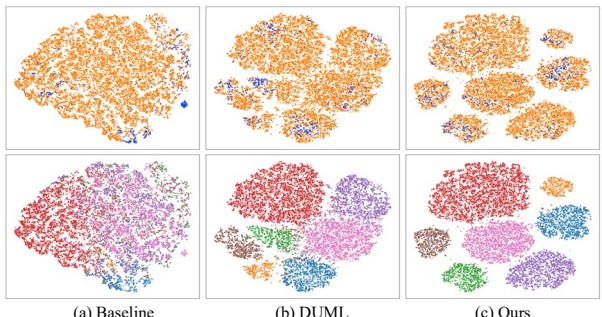

**Fig. 7. T-SNE visualizations of feature embeddings on the '→ CK+' task. The first row denotes domain information (Orange: source domain; Blue: target domain) while the second row represents category information (Each color denotes a class).**

noticeable mismatch with the source domain. With domain-uncertainty estimations, DUML better aligns the source domains to the target one but exhibits relatively poor intra-class compactness where the decision boundaries are not clear enough. In contrast, our LA-CMFER surpasses both Baseline and DUML, as evidenced by the formation of clusters with more distinct boundaries. This signifies the superior transferability of our approach which can effectively maintain the strong discrimination capability.

*Notably, more additional experiments including 'Selections of Reliability Threshold $\epsilon$', 'Number of Source Domains', and so on are given in the Supplementary Materials.*

## 5 Conclusion

In this paper, we present the LA-CMFER framework to achieve the CMFER task by addressing both inter- and intra-domain shifts. We first propose a dual-level (i.e., sample- and cluster-level) inter-domain alignment method to narrow the inter-domain shifts by prioritizing hard-to-align samples and sufficiently using the class attributes. Then, to tackle the intra-domain shifts, a multi-view intra-domain alignment method is introduced to promote consistency between the global and local branches. Extensive experiments have verified the superiority of our LA-CMFER.

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
