# OpenReview forum: "Learning with Alignments: Tackling the Inter- and Intra-domain Shifts for Cross-multidomain Facial Expression Recognition"
_acmmm.org/ACMMM/2024/Conference — MM2024 Poster_

### Official Review · Reviewer_tZok · 2024-05-20

**Rating:** 4
**Confidence:** 4

**Summary:**

This paper proposes the LA-CMFER method to address cross-multidomain in facial expressions. By handling both inter-domain and intra-domain transitions, it achieves effective CMFER. Experiments validate the effectiveness of the method.

**Strengths:**

1. The proposed LA-CMFER method effectively addresses the cross-multidomain challenges in facial expression recognition.
2. Extensive experiments have validated its effectiveness.

**Limitations:**

1. In FER, capturing subtle changes in facial expressions is crucial. The authors' focus on both local and global aspects of the face is reasonable. However, why does dividing the face into 4 fixed regions ensure the capture of subtle facial movements? Generally, facial images have different poses, making it difficult for 4 fixed local patches to consistently focus on the same facial subregion.
2. How does MPC work? The paper does not provide a detailed explanation of Eq.10.
3. Given that the proposed LA-CMFER and DUML (ACM MM’23) both address the CMFER task, please clarify the differences between the two methods.
4. Please provide a deeper discussion of the proposed method, including its 'limitations'.
5. Open-source the code.
6. The paper's descriptions need serious improvement, especially the explanation of equations such as Eq.1 and Eq.3.

**Suitability:**

2

---

### Official Review · Reviewer_1H9a · 2024-05-21

**Rating:** 4
**Confidence:** 3

**Summary:**

This paper studied the cross-multidomain problems with inter- and intra-domain shifts in facial expression recognition. They introduced a dual-level inter-domain alignment method to prioritize hard-to-align samples and foster a well-clustered feature space. Meanwhile, they also designed a muti-view clustering consistency constraint between the global and local branches to gain a concentrated target feature space for alleviating the interference of noisy samples.

**Strengths:**

This paper provides a novel Learning with Alignments for Cross-Multidomain Facial Expression Recognition (LA-CMFER) framework to simultaneously tackle inter- and intra-domain shifts. To address the inter-domain shifts, they use the scaling strategy to pay more attention and give higher weight to those hard-to-align samples with higher uncertainty between different domains and generate a more well-clustered feature space. As for the intra-domain shifts, they utilized a multi-view voting scheme to gain high-quality pseudo labels for an effective information interaction in the dual-level branches. The methodology in this paper looks professional, it can be seen that the theory of this paper is very solid.

**Limitations:**

There are some problems, which must be solved before it is considered for publication. If the following problems are well-addressed, the reviewer believes that the essential contribution of this paper is important for facial expression recognition.

1.  **Inconsistent Data**: In **Tabel 1**, the experimental data of **DMUL** is not consistent in their original paper[1]. It is not known whether this inconsistency is due to an error while copying or other reasons. However, DUML is a method to solve the same problem in CMFER, and we **suggest** that the results must keep correct and objective.


2.  **Inproper tuning**: In Equation (14), we can find that the final loss function consists of four parts, related to three hyper-parameters (i.e. $\alpha$, $\beta$, and $\gamma$). However,  in **Section 4.3** , we find that the sum of hyper-parameters(i.e. $\alpha$, $\beta$, and $\gamma$) of the loss function is not consistent for different target datasets (for the '→CK+' task, $\alpha$ + $\beta$ + $\gamma$ = 0.7, while for the '→FER- 2013' task, $\alpha$ + $\beta$ + $\gamma$ = 1.0). It seems that the authors did not consider the influence of these loss functions among each other, but tuned them separately. It is hard to convince us that the parameter-sensitivity experiments is proper and can strongly support your methods. The **suggestion** is that you need to consider the effects of different loss with each other and reset the hyper-parameter ablation experiments.


3.  **Complexity Analysis**: Following the paper's work, this structure may have some effectiveness in solving the cross-multidomain FER, but it also increases the complexity of the framework compared with traditional MDA methods in FER. We **advise** you to add the comparison analysis on computational costs and training time with other SOTA methods.


* * *
**Reference:**

[1] Liu H, Cai H, Lin Q, et al. Learning from More: Combating Uncertainty Cross-multidomain for Facial Expression Recognition[C]//Proceedings of the 31st ACM International Conference on Multimedia. 2023: 5889-5898.

**Suitability:**

2

---

### Official Review · Reviewer_Qann · 2024-05-21

**Rating:** 5
**Confidence:** 3

**Summary:**

This paper proposed an method for multi-source domain adaptive facial expression recognition. In this paper, inter-domain shift means the diverse between a source dataset and a target dataset, while intra-domain shift means the diverse between different classes in the target dataset. The proposed method designed sample- and cluster-level alignment loss to handle the inter-domain shift, and a multi-view (local and global features) voting scheme for minimizing the intra-domain shift. The results are SOTA.

**Strengths:**

1. This paper is well organized and easy to follow.

**Limitations:**

1.The words ‘cross-multidomain’ may be not accurate enough. This paper tries to transfer the emotional Knowledge from multiple source domains (datasets) to a single target domain (dataset). In domain adaptation area, this task is called multi-source domain adaptive FER.

2.The definition of domain is not clear. The meaning of 'domain' in this paper is more more like 'dataset'. In this paper, inter-domain shift is actually cross-dataset domain shift, and intra-domain shift is intra-dataset domain shift. Moreover, The images in AffectNet is diverse, which may contain data from different domains.  It will be better if the author describe this point more accurate.

3.It is common to concurrently use the local and global features for (cross-domain) FER. However, what is the motivation of using both of them in this paper? and what is new compared to published methods that based on local and global features?

4.The idea of ‘sample-level inter-domain alignment’ is very similar with that of ECAN [1], which also reweight samples and use MMD.

5.I find the ‘intra-domain’ shift is handled on the target dataset. Is there ‘intra-domain’ shift in a source dataset?

[1] A Deeper Look at Facial Expression Dataset Bia, IEEE TAFFC, 2020.

**Suitability:**

2

---

### Official Review · Reviewer_t9gq · 2024-05-23

**Rating:** 4
**Confidence:** 2

**Summary:**

This paper proposes a Learning with Alignments for Cross-Multidomain Facial Expression Recognition (LA-CMFER) framework to tackle the inter - and intra -domain shifts. To this end, authors use a dual-level alignment and a multi-view alignment for inter-domain and intra-domain shifts, respectively.

**Strengths:**

1. The paper is well-organized and clearly written.
2. The experiments are extensive and helpful to validate the effectiveness of the model.

**Limitations:**

1. How do you ensure that the manipulation of emphasizing hard sample alignment are optimally performed to maintain a balance between removing noise from multiple labeled source domains and preserving relevant information according to decision boundaries.
2. To display the shapes and colors in the Fig.2 more clearly, the authors are suggested to add the descriptions.
3. The conclusion section is a little short, which should more explicitly summarize the main contributions.

**Suitability:**

3

---

### Meta-Review · Area_Chair_qrHG · 2024-07-04

**Recommendation:** Accept (Poster)
**Confidence:** 5

**Metareview:**

The authors made a strong case in the rebuttal and most reviewers are satisfied with the responses and clarifications provided. From the technical standpoint, the work introduces significant strides into cross-multidomain facial expression recognition by tackling the problem of domain shift from across multiple domains and internally within specific domains $-$ this provides useful insights to the research community working on this topic. The validation of the proposed method was very comprehensive and was carried out on six benchmark datasets, showcasing its robustness and generalizability. The ablation studies provided were also comprehensive, but appear quite mechanical (as with many papers these days). There wasn't a sufficient investigation into the intricacies of the alignment process, which was reduced to merely loss functions that are optimized together with other loss functions. I thought the number of source domains would be something that warrants attention in the main content of the paper but was relegated to the supplementary material. It is crucial to understand how this alignment would be impacted when the number of domains increases since optimizing across domains will be more challenging. All things considered, this is a borderline case but if possible, I would recommend accepting this paper as a poster at ACM MM.

Two points to note:
* Despite some reviewers indicating that this work is "multimodal" in nature, I disagree. The notion of "domain" is not equivalent to a "modality". It is obvious that the modality of interest in this work is image, and _only_ image (note: CK+ is from video but individual extracted frames are typically used). These images can be obtained from a variety of "domains" (separately collected datasets), as what this paper looks into. As the theme indicates priority for truly multi-modal works in ACM MM, this work may be slightly disadvantaged despite the strong reviews received. As this work is also suitable for affective computing or machine learning venues, I am not strongly against not accepting this paper.
* The original submitted format of the paper is not ACM's and the authors should have it formatted to the standard ACM paper format. It is a very simple instruction to follow.